# Multi-study fMRI outlooks on subcortical BOLD responses in the stop-signal paradigm

**Scott Isherwood[1]\*[†], Sarah A Kemp[1,2][†], Steven Miletić[1,3], Niek Stevenson[1], Pierre-Louis Bazin[4], Birte Forstmann[1]**

[1]Integrative Model-Based Cognitive Neuroscience Research Unit, University of Amsterdam, Amsterdam, Netherlands; [2]Sensorimotor Neuroscience and Ageing Research Lab, School of Psychological Sciences, University of Tasmania, Hobart, Australia; [3]Department of Psychology, Faculty of Social Sciences, Leiden University, Leiden, Netherlands; [4]Full brain picture Analytics, Leiden, Netherlands

## eLife Assessment

This study aggregates across five fMRI datasets and reports that a network of brain areas previously associated with response inhibition processes, including several in the basal ganglia, are more active on failed stop than successful stop trials. This study is **valuable** as a well-powered investigation of fMRI measures of stopping, and following revisions provides **solid** evidence for its conclusions.

**\*For correspondence:**
scott@leeclan.net

[†]These authors contributed equally to this work

**Competing interest:** The authors declare that no competing interests exist.

**Abstract** This study investigates the functional network underlying response inhibition in the human brain, particularly the role of the basal ganglia in successful action cancellation. Functional magnetic resonance imaging (fMRI) approaches have frequently used the stop-signal task to examine this network. We merge five such datasets, using a novel aggregatory method allowing the unification of raw fMRI data across sites. This meta-analysis, along with other recent aggregatory fMRI studies, does not find evidence for the innervation of the *hyperdirect* or *indirect* cortico-basal-ganglia pathways in successful response inhibition. What we do find, is large subcortical activity profiles for *failed stop* trials. We discuss possible explanations for the mismatch of findings between the fMRI results presented here and results from other research modalities that have implicated nodes of the basal ganglia in successful inhibition. We also highlight the substantial effect smoothing can have on the conclusions drawn from task-specific general linear models. First and foremost, this study presents a proof of concept for meta-analytical methods that enable the merging of extensive, unprocessed, or unreduced datasets. It demonstrates the significant potential that open-access data sharing can offer to the research community. With an increasing number of datasets being shared publicly, researchers will have the ability to conduct meta-analyses on more than just summary data.

## Introduction

Response inhibition, generally defined as the ability to suppress a planned or already-initiated response (*Logan, 1985*), is an essential part of everyday motor control, and underpinned by a series of cortical and subcortical pathways. Defining the neural mechanisms underlying response inhibition in the neurotypical population has important consequences in the clinical neurosciences, where impairment in these pathways has been associated with a number of neurological and psychiatric diseases including Parkinson's disease, addiction, and schizophrenia (*Chowdhury et al., 2018*; *Claassen et al., 2015*; *Congdon et al., 2014*; *Noël et al., 2016*; *Rømer Thomsen et al., 2018*; *Seeley et al., 2009*).

Response inhibition has been behaviourally examined using the stop-signal task (SST) for more than four decades. In the SST, participants make a motor response as quickly as possible in response to a go signal. In a minority of trials (usually around 25% of all trials), a stop signal appears shortly after the onset of the go signal, indicating that the participant should not respond to the go signal in that trial. The stop signal's onset is normally adjusted after each stop-signal trial based on stopping success, such that each participant will be able to stop successfully on approximately 50% of trials (*Verbruggen et al., 2019*). Behavioural dynamics during the SST are interpreted under the framework of the horse-race model (*Logan and Cowan, 1984*). This model proposes that on each stop trial, the presentation of the go stimulus triggers the go process, which races towards a threshold that results in a response. Upon the presentation of the stop signal, a stop process is similarly triggered, which races towards an independent threshold. Depending on whether the go or stop process finishes first, the response is, respectively, performed or inhibited. Performance on go trials and failed stop trials (where the participant makes an inappropriate response) is quantified by reaction time (RT). Inhibition performance in the SST as a whole is quantified by the stop-signal reaction time (SSRT), which estimates the speed of the latent stopping process (*Verbruggen et al., 2019*).

Contemporary models of response inhibition propose that inhibition is realized via three cortico-basal-ganglia pathways; the *direct*, *indirect*, and *hyperdirect* pathways (*Diesburg and Wessel, 2021*; *Mink, 1996*). While all three are involved in response inhibition and movement, the *hyperdirect* pathway has been theorized to be the pathway through which action is ultimately cancelled (*Aron and Poldrack, 2006*). The signalling cascade originates from the prefrontal cortex and is thought to implement stopping upon detection of a stop signal by inhibition of the thalamus (Tha) via the subthalamic nucleus (STN), substantia nigra (SN), and globus pallidus interna (GPi; *Coudé et al., 2018*; *Diesburg and Wessel, 2021*). This pathway was originally identified in rodents and non-human primates (*Eagle et al., 2008*; *Schmidt et al., 2013*), but its anatomical plausibility in humans was demonstrated by Chen et al., who measured firing in the frontal cortex 1–2 ms after stimulation of the STN (*Chen et al., 2020*). The connectivity of these cortico-basal-ganglia tracts has been shown to be correlated with stopping behaviour (*Forstmann et al., 2012*; *Singh et al., 2021*; *Xu et al., 2016*; *Zhang and Iwaki, 2020*). Clinical studies have also demonstrated the importance of subcortical regions, particularly the STN, in relation to stopping. A multitude of these studies provide electrophysiological support for the involvement of the STN in successful response inhibition (*Alegre et al., 2013*; *Benis et al., 2014*; *Benis et al., 2016*; *Fischer et al., 2017*; *Mosher et al., 2021*; *Wessel et al., 2016*), indicating that increased β-band activity induces global motor suppression. Evidence from Parkinson's disease patients undergoing deep brain stimulation has also associated the STN with (successful) stopping behaviour (*Mirabella et al., 2012*; *Ray et al., 2009*; *Ray et al., 2012*; *Swann et al., 2011*; *van den Wildenberg et al., 2021*) and demonstrated that bilateral stimulation of this region can improve performance in the SST (*Mancini et al., 2018*).

Functional imaging research has also been used extensively to elucidate which regions are associated with response inhibition. These images are frequently acquired at 3 Tesla (T) and the BOLD responses interpreted by the use of contrast analyses, subtracting the activity of regions during different conditions. In the SST, these conditions are *go* (*GO*), *failed stop* (*FS*), and *successful stop* (*SS*) trials. Contrasts of interest are often *FS > GO*, *SS > GO*, and *FS > SS*. Cortically, three regions have been consistently implicated in successful inhibition: the right inferior frontal gyrus (rIFG), pre-supplementary motor area (preSMA), and anterior insula (*Aron et al., 2014*; *de Hollander et al., 2017*; *Isherwood et al., 2023*; *Miletić et al., 2020*; *Swick et al., 2011*). In the subcortex, functional evidence is relatively inconsistent. Some studies have found an increase in BOLD response in the STN in SS > GO contrasts (*Aron and Poldrack, 2006*; *Coxon et al., 2016*; *Gaillard et al., 2020*; *Yoon et al., 2019*), but others have failed to replicate this (*Bloemendaal et al., 2016*; *Boehler et al., 2010*; *Chang et al., 2020*; *Xu et al., 2015*). Moreover, some studies have actually found higher STN, SN, and thalamic activation in failed stop trials, not successful ones (*de Hollander et al., 2017*; *Isherwood et al., 2023*; *Miletić et al., 2020*).

Here, we reprocess and reanalyse five functional SST datasets to shed light on the discrepancies in subcortical BOLD responses. Canonical methods of meta-analysis have the tendency to lose information when compiling multiple sources of data, due to reliance on summary statistics and a lack of raw data accessibility. Taking advantage of the recent surge in open-access data, we aimed to improve upon these methods by using the raw data now available instead of relying on simple

**Table 1.** Descriptive statistics of behaviour in the SST across each dataset. Standard errors are given.

| Dataset | Median go RT (ms) | Median failed stop RT (ms) | Go omissions (%) | Go errors (%) | Mean SSRT (ms) | Median SSD (ms) | Mean stopping accuracy (%) |
|---|---|---|---|---|---|---|---|
| *Aron_3T* | 423 ± 18 | 382 ± 11 | 0.7 ± 0.4 | 0.6 ± 0.2 | 189 ± 8 | 227 ± 17 | 53 ± 1 |
| *Poldrack_3T* | 466 ± 9 | 426 ± 8 | 0.1 ± 0.04 | 0.9 ± 0.1 | 209 ± 5 | 279 ± 11 | 52 ± .6 |
| *deHollander_7T* | 472 ± 24 | 439 ± 22 | 1.6 ± 0.5 | 0.3 ± 0.1 | 219 ± 8 | 250 ± 22 | 54 ± 2 |
| *Isherwood_7T* | 626 ± 25 | 543 ± 22 | 1.9 ± 0.4 | 2.2 ± 0.4 | 256 ± 8 | 350 ± 30 | 54 ± 1 |
| *Miletic_7T* | 445 ± 17 | 414 ± 15 | 1.1 ± 0.5 | 0.7 ± 0.2 | 219 ± 20 | 230 ± 23 | 50 ± 1 |

summary measures (e.g., MNI coordinates). Though computationally expensive, the gain in power from reanalysing multiple functional datasets without this loss of information is of huge benefit. In addition, using raw data as a starting point for datasets acquired separately allows one to minimize differences in preprocessing and analyses pipelines. We chose datasets that used similar go stimuli (left or right pointing arrows) to maintain as much consistency across the datasets as possible. Stop signals during the SST are generally either of the auditory or visual type; we opted to use both types in this study with the assumption that they rely on the same underlying inhibition network (*Ramautar et al., 2006*).

## Results

### Behavioural analyses

*Table 1* summarizes the descriptive statistics of the behavioural data from each dataset. Consistent with the assumptions of the standard horse-race model (*Logan and Cowan, 1984*), the median failed stop RT is significantly faster within all datasets than the median go RT (*Aron_3T*: p < 0.001, $BF_{log10}$ = 2.77; *Poldrack_3T*: p < 0.001, $BF_{log10}$ = 23.49; *deHollander_7T*: p < 0.001, $BF_{log10}$ = 8.88; *Isherwood_7T*: p < 0.001, $BF_{log10}$ = 2.95; *Miletic_7T*: p = 0.0019, $BF_{log10}$ = 1.35). Mean SSRTs were calculated using the integration method and are all within normal range across the datasets. The mean stopping accuracy (near 50%) across all datasets indicates that the staircasing procedure operated accordingly and successfully kept stop-signal delays (SSDs) tailored to the SSRT of participants during the task. Longer RTs were found in the *Isherwood_7T* dataset in comparison to the four other datasets. The only difference in procedure in the *Isherwood_7T* dataset is the use of a visual stop signal as opposed to an auditory stop signal. This RT difference is consistent with previous research, where auditory stop signals and visual go stimuli have been associated with faster RTs compared to unimodal visual presentation (*Carrillo-de-la-Peña et al., 2019*; *Weber et al., 2024*). The mean SSRTs and probability of stopping are within normal range, indicating that participants understood the task and responded in the expected manner.

To observe quantitative differences in signal quality between the datasets, we first calculated region of interest (ROI)-wise temporal signal-to-noise ratio (tSNR) maps of the unsmoothed data. In *Figure 1*, we show both the corrected and uncorrected tSNR values for five ROIs. As the tSNR values across each hemisphere were similar, we opted to take the mean across both. The corrected tSNR values display the clear benefit of 7T acquisition compared to 3T in terms of data quality. In the cortical ROIs, the 7T datasets appear to perform equally well, though when zooming in on subcortical ROIs, the *deHollander_7T* and *Miletic_7T* datasets display superiority. The uncorrected tSNR values paint a different picture. These tSNRs are even across all the datasets, with the exception of the Isherwood_7T dataset which appears to suffer, most likely due to its increased multiband factor (*Chen et al., 2015*). It should be noted that interpretation of the uncorrected tSNR values is difficult, due to the inherent proportionality of tSNR and voxel volume (*Edelstein et al., 1986*). That is, the 3T datasets acquire data with a voxel volume approximately 10 times smaller than that of the 7T datasets and therefore have an advantage when not correcting for this difference.

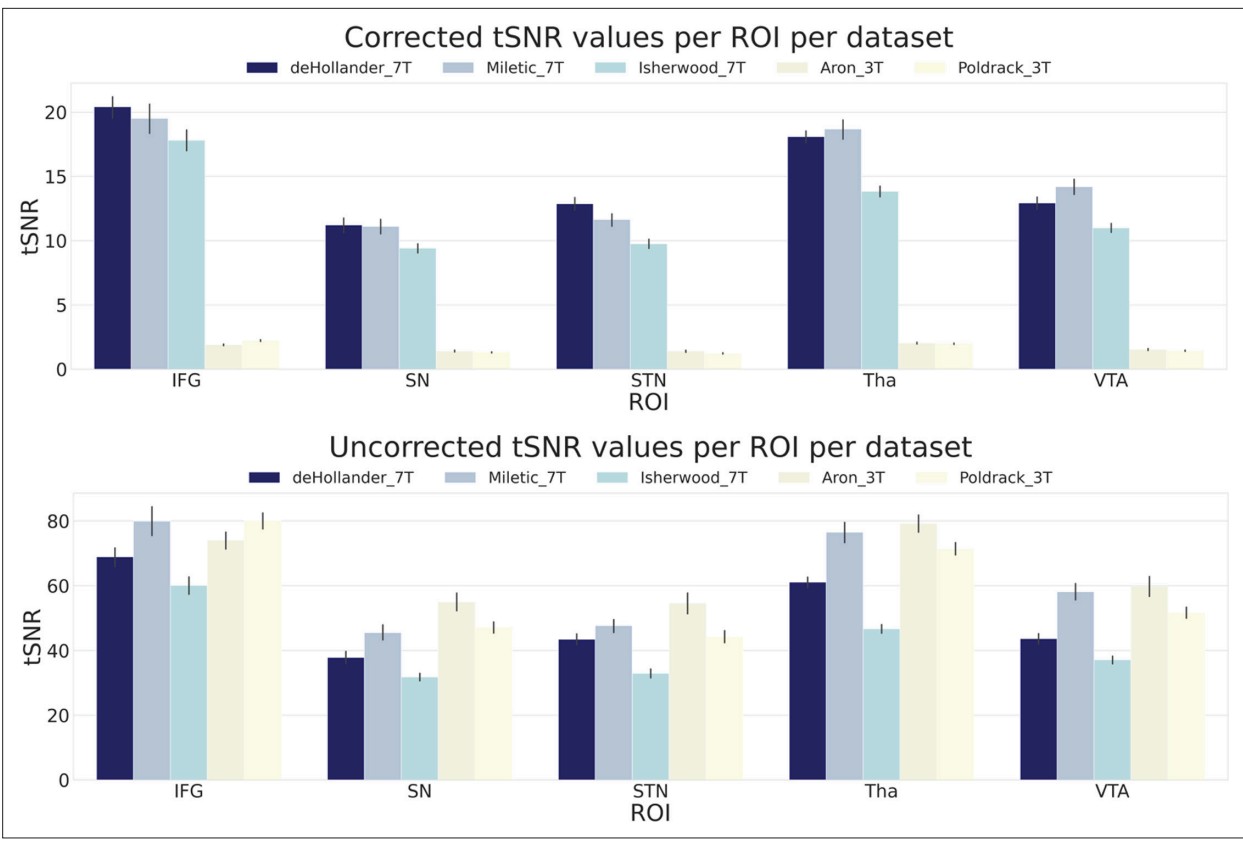

**Figure 1.** Corrected and uncorrected temporal signal-to-noise ratio (tSNR) values for five regions of interest (ROIs) over all datasets. The values are derived from the mean tSNR values of both hemispheres. Error bars are standard errors. Corrected tSNRs are equal to the uncorrected tSNRs divided by the volume of a single voxel. IFG, inferior frontal gyrus; SN, substantia nigra; STN, subthalamic nucleus; Tha, thalamus; VTA, ventral tegmental area.

## Voxel-wise general linear models

We calculated whole-brain voxel-wise general linear models (GLMs) using the canonical hemodynamic response function (HRF) with a temporal derivative to statistically test the brain areas underlying behaviour in the SST. The three trial types result in three possible contrasts: FS > GO, FS > SS, and SS > GO. Due to the restricted field of view (FOV) of the images acquired in the *deHollander_7T* dataset, group-level statistical parametric maps (SPMs) display a limited activation pattern at the most superior part of the cortex, as no data were acquired there for one dataset. We first show the group-level SPMs of the overall contrasts of the SST across all datasets (see *Figure 2*), the SPMs for each contrast of each individual dataset can be found in *Figure 2—figure supplements 1–3*. See *Figure 2—figure supplement 4* for the group analyses where the FS and SS trials were time-locked to the stop-signal onset. Significant BOLD responses for the FS > GO contrast were found in the bilateral IFG, preSMA, SN, STN, and ventral tegmental area (VTA). It can be clearly seen that this contrast elicits the largest subcortical response out of the three. The FS > SS contrast shows significant bilateral activation in the IFG, STN, Tha, and VTA. The SS > GO contrast shows significant activation in the bilateral IFG and Tha.

## ROI-wise GLMs

To further statistically compare the functional results between datasets, we then fit a set of GLMs using the canonical HRF with a temporal derivative to the timeseries extracted from each ROI. Below we show the results of the group-level ROI analyses over all datasets using *z*-scores (*Figure 3*) and log-transformed Bayes Factors (BFs; *Figure 4*). Note that these values were time-locked to the onset of the go signal. See *Figure 3—figure supplement 1* for analyses where the FS and SS trials were time-locked to the onset of the stop signal. To account for multiple comparisons, threshold values were set using the false discovery rate (FDR) method for the frequentist analyses.

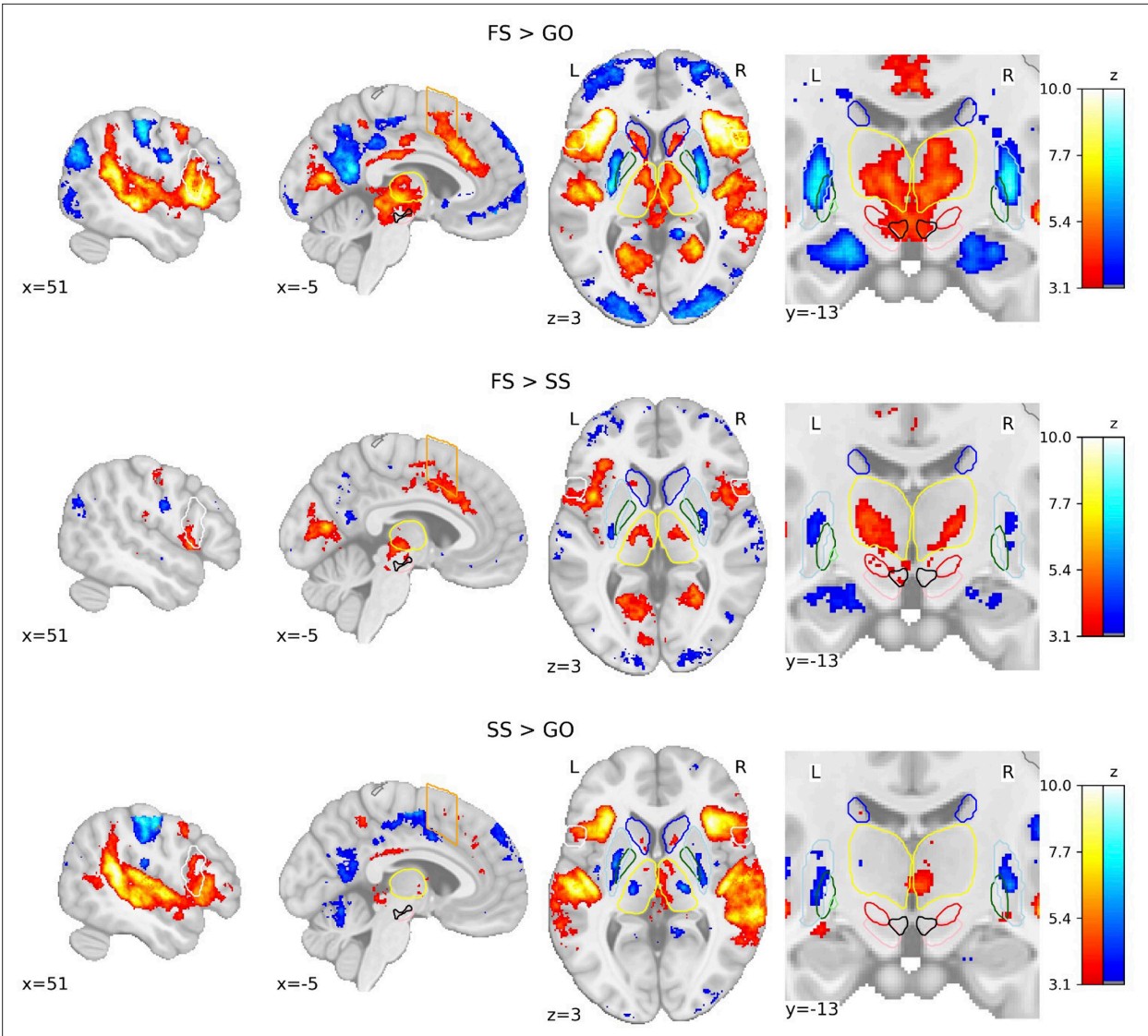

**Figure 2.** Group-level statistical parametric maps (SPMs) of the three main contrasts of the stop-signal task (SST). Activation colours indicate false discovery rate (FDR) thresholded (*q* < 0.05) *z*-values. Two sagittal, one axial, and one zoomed in coronal view are shown. Coloured contour lines indicate regions of interest (inferior frontal gyrus [IFG] in white, M1 in grey, pre-supplementary motor area [preSMA] in orange, Caudate in dark blue, Putamen in light blue, GPe in dark green, globus pallidus interna [GPi] in light green, substantia nigra [SN] in pink, subthalamic nucleus [STN] in red, thalamus in yellow, and ventral tegmental area [VTA] in black). The background template and coordinates are in MNI2009c (1 mm). FS, failed stop; SS, successful stop.

The online version of this article includes the following figure supplement(s) for figure 2:

**Figure supplement 1.** Group-level statistical parametric maps (SPMs) of the FS > GO contrast of the stop-signal task (SST) for each dataset.

**Figure supplement 2.** Group-level statistical parametric maps (SPMs) of the FS > SS contrast of the stop-signal task (SST) for each dataset.

**Figure supplement 3.** Group-level statistical parametric maps (SPMs) of the SS > GO contrast of the stop-signal task (SST) for each dataset.

**Figure supplement 4.** Group-level statistical parametric maps (SPMs) of the three main contrasts of the stop-signal task (SST), where SS and FS trials were time-locked to the presentation of the stop signal.

For the FS > GO contrast, the frequentist analysis found significant positive *z*-scores in all regions bar left and right M1, and the left GPi. The right M1 showed a significant negative *z*-score; left M1 and GPi showed no significant effect in this contrast. The BFs showed moderate or greater evidence for the alternative hypothesis in bilateral IFG, preSMA, caudate, STN, Tha, and VTA, and right GPe. Bilateral M1 and left GPi showed moderate evidence for the null. Evidence for other ROIs was anecdotal (see *Figure 4*). For the FS > SS contrast, we found significant positive *z*-scores in in all regions except

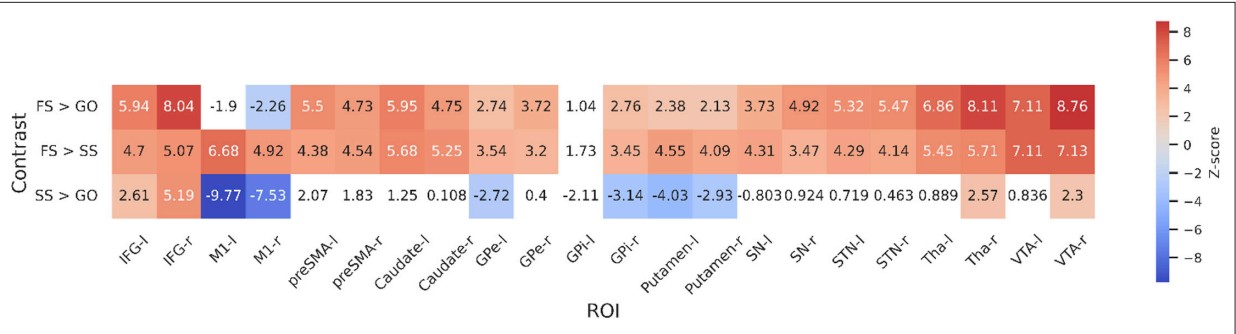

**Figure 3.** Group-level *z*-scores from the region of interest (ROI)-wise general linear model (GLM) analysis of included datasets. Thresholds are set using false discovery rate (FDR) correction (*q* < 0.05), varying between contrasts. The thresholds for each contrast are as follows: 3.01 for FS > GO, 2.26 for FS > SS, and 3.1 for SS > GO. Regions that do not reach significance are coloured white. Left and right hemispheres are shown separately, denoted by '-l' or '-r', respectively. IFG, inferior frontal gyrus; M1, primary motor cortex; preSMA, pre-supplementary motor area; GPe, globus pallidus externa; GPi, globus pallidus interna; SN, substantia nigra; STN, subthalamic nucleus; Tha, thalamus; VTA, ventral tegmental area.

The online version of this article includes the following figure supplement(s) for figure 3:

**Figure supplement 1.** Group-level *z*-scores from the region of interest (ROI)-wise general linear model (GLM) analysis of included datasets, where SS and FS trials were time-locked to the presentation of the stop signal.

**Figure supplement 2.** A comparison of the BFs and the frequentist *z*-scores from FSL.

the left GPi. The BFs showed moderate or greater evidence for right IFG, right GPi, and bilateral M1, preSMA, Tha, and VTA, and moderate evidence for the null in left GPi. Evidence for other ROIs was anecdotal (see *Figure 4*). For the SS > GO contrast we found a significant positive *z*-scores in bilateral IFG, right Tha, and right VTA, and significant negative *z*-scores in bilateral M1, left GPe, right GPi, and bilateral putamen. The BFs showed moderate or greater evidence for the alternative hypothesis in bilateral M1 and right IFG, and moderate or greater evidence for the null in left preSMA, bilateral caudate, bilateral GPe, left GPi, bilateral putamen, and bilateral SN. Evidence for other ROIs was anecdotal (see *Figure 4*).

Although the frequentist and Bayesian analyses are mostly in line with one another, they do detect some differences, particularly in the contrasts with FS. In the FS > GO contrast, the interpretation of the GPi, GPe, putamen, and SN differ. The frequentist models suggest significantly increased activation for these regions (bar left GPi) in FS trials. In the Bayesian model, this evidence was found to be anecdotal in the SN and right GPi, and moderate in the right GPe, while finding anecdotal or moderate evidence for the *null* hypothesis in the left GPe, left GPi, and putamen. For the FS > SS contrast, the frequentist analysis found significant activation in all regions except for the left GPi, whereas the Bayesian analysis found this evidence to be only anecdotal, or in favour of the null for

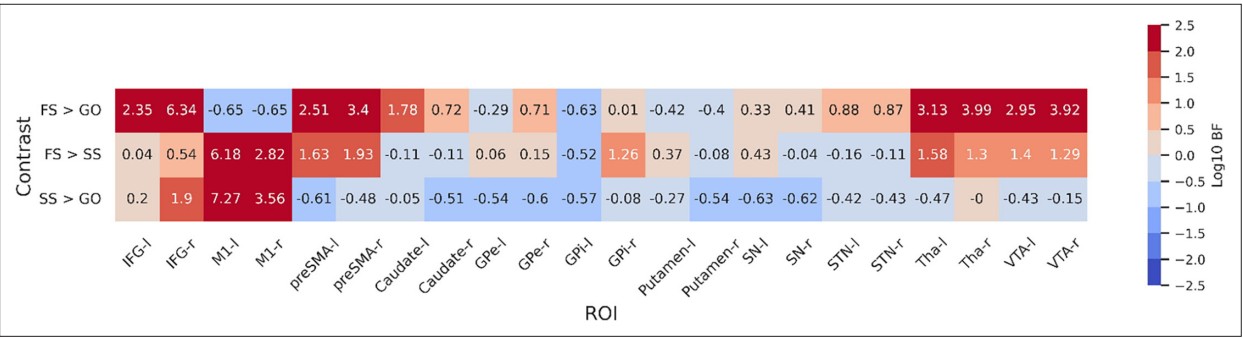

**Figure 4.** Log-transformed Bayes Factors for each contrast based on parameter estimates from first-level model. Colouring depicts evidence for each hypothesis based on a variation of Jeffreys' scale. BFs more than 2 or less than 2 on a log scale are defined as extreme evidence. Refer to *Table 4* for details on interpretation of log-transformed BFs. BFs were calculated for each contrast for both hemispheres of each region of interest (ROI). Left and right hemispheres are shown separately, denoted by '-l' or '-r', respectively. IFG, inferior frontal gyrus; M1, primary motor cortex; preSMA, pre-supplementary motor area; GPe, globus pallidus externa; GPi, globus pallidus interna; SN, substantia nigra; STN, subthalamic nucleus; Tha, thalamus; VTA, ventral tegmental area.

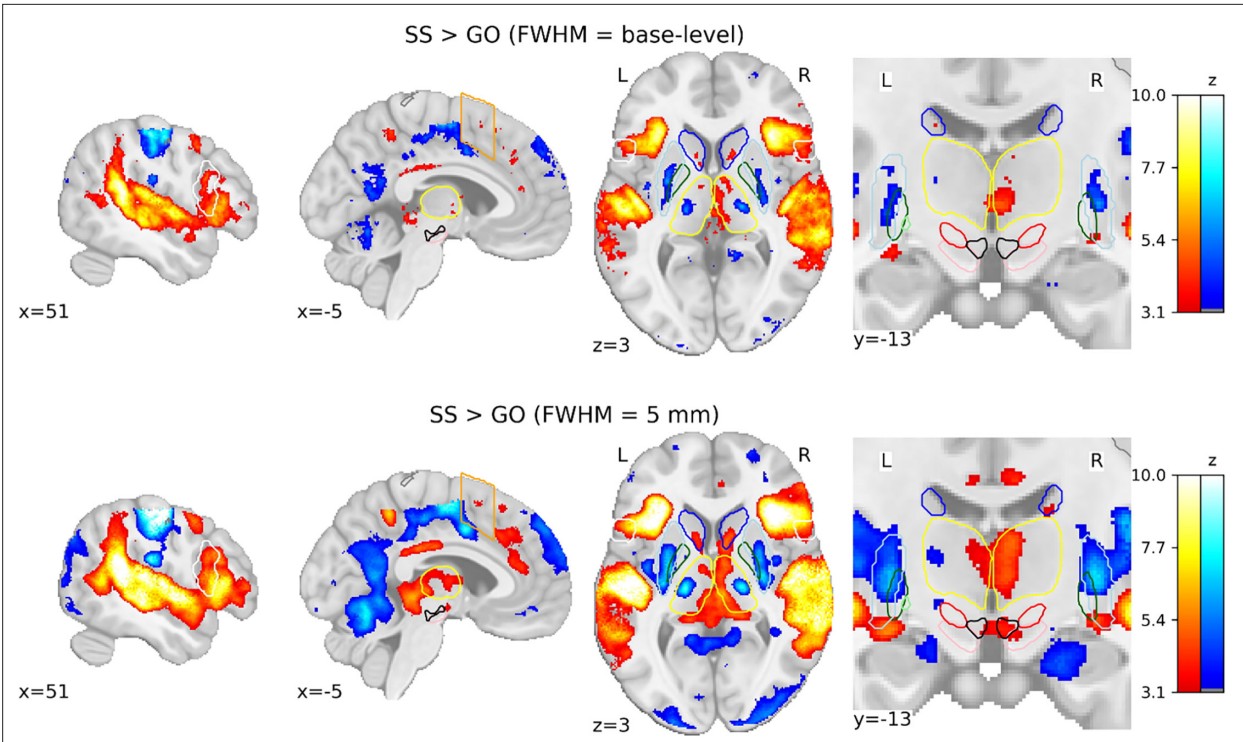

**Figure 5.** Comparison of group-level statistical parametric maps (SPMs) for the SS > GO contrast using different smoothing kernels. SPMs resulting from general linear models (GLMs) computed on base-level spatially smoothed data can be seen on the top row, with SPMs resulting from GLMs computed on data spatially smoothed with a full width half maximum (FWHM) of 5 mm. Activation colours indicate false discovery rate (FDR) thresholded ($q < 0.05$) $z$-values. Two sagittal, one axial, and one zoomed in coronal view are shown. Coloured contour lines indicate regions of interest (inferior frontal gyrus [IFG] in white, M1 in grey, pre-supplementary motor area [preSMA] in orange, Caudate in dark blue, Putamen in light blue, GPe in dark green, globus pallidus interna [GPi] in light green, substantia nigra [SN] in pink, subthalamic nucleus [STN] in red, thalamus in yellow, and ventral tegmental area [VTA] in black). The background template and coordinates are in MNI2009c (1 mm). FS, failed stop; SS, successful stop.

The online version of this article includes the following figure supplement(s) for figure 5:

**Figure supplement 1.** Group-level statistical parametric maps (SPMs) of the FS > GO and FS > SS contrasts using different smoothing kernels.

a large number of regions (see *Figure 4* for details; note that *Figure 3*) shows *z*-scores, thus more extreme values indicate an effect in that direction. In contrast, *Figure 4* shows log BFs, and thus positive values indicated support of an effect in *any* direction. *Figure 3—figure supplement 2* shows a comparison between the *z*-scores and the BFs.

## Smoothing comparison

To visualize the effect of spatial smoothing on voxel-wise GLMs, we computed SST contrasts using base-level kernels and a kernel of 5 mm. The difference in group-level SPMs for SS > GO contrast is prominent (see *Figure 5*). Comparisons for the contrasts of FS > GO and FS > SS contrasts can be found in *Figure 5—figure supplement 1*. If we were to make inferences based on the group-level SPMs calculated using the 5 mm kernel, this study could potentially conclude that both the SN and VTA are significantly activated in SS trials compared to GO trials. Much larger regions of significant activation can be seen in the 5 mm smoothed SPMs, both cortically and subcortically. This comparison demonstrates the prominent consequences that preprocessing pipelines can have on the overall analysis of functional data.

## Discussion

The functional network underlying response inhibition in the human brain has been a key research question in the cognitive neurosciences for decades. The basal ganglia specifically have been implicated in broad movement control since the early twentieth century (*Wilson, 1912*). However, the

precise role of these subcortical structures in successful response inhibition is still unclear. Evidence for the role of the basal ganglia in response inhibition comes from a multitude of studies citing significant activation of either the SN, STN, or GPe during successful inhibition trials (*Aron, 2007*; *Aron and Poldrack, 2006*; *Mallet et al., 2016*; *Nambu et al., 2002*; *Zhang and Iwaki, 2019*). Here, we re-examined activation patterns in the subcortex across five different datasets, identifying differences in regional activation using both frequentist and Bayesian approaches. Broadly, the frequentist approach found significant differences between most ROIs in FS > GO and FS > SS contrasts, and limited differences in the SS > GO contrast. The Bayesian results were more conservative; while many of the ROIs showed moderate or strong evidence, some with small but significant *z*-scores were considered only anecdotal by the Bayesian analysis. In our discussion, where the findings between analytical approaches differ, we focus mainly on the more conservative Bayesian analysis.

Here, our multi-study results found limited evidence that the canonical inhibition pathways (the indirect and hyperdirect pathways) are recruited during successful response inhibition in the SST. We expected to find increased activation in the nodes of the indirect pathway (e.g., the preSMA, GPe, STN, SN, GPi, and thalamus; *Diesburg and Wessel, 2021*) during successful stop compared to go or failed stop trials. We found strong evidence for activation pattern differences in the preSMA, thalamus, and right GPi between the two stop types (failed and successful), and limited evidence, or evidence in favour of the null hypothesis, in the other regions, such as the GPe, STN, and SN. However, we did find recruitment of subcortical nodes (VTA, thalamus, STN, and caudate), as well as preSMA and IFG activation during *failed* stop trials. We suggest that these results indicate that failing to inhibit one's action is a larger driver of the utilization of these nodes than action cancellation itself.

These results are in contention to many previous functional magnetic resonance imaging (fMRI) studies of the SST as well as research using other measurement techniques such as local field potential recordings, direct subcortical stimulation, and animal studies, where activation of particularly the STN has consistently been observed (*Aron and Poldrack, 2006*; *Benis et al., 2014*; *Fischer et al., 2017*; *Mancini et al., 2018*; *Wessel et al., 2016*). Attempting to reconcile these discrepancies leaves us with a few discussion points. The first is that of the reliability of studying the subcortex in vivo using fMRI. Due to its distance from the MR head coils, proximity of subregions and varying biophysical properties, the subcortex can have limited inter-regional contrast and a low signal-to-noise ratio (*Bazin et al., 2020*; *de Hollander et al., 2017*; *Isaacs et al., 2018*; *Isherwood et al., 2021a*; *Keuken et al., 2018*; *Miletić et al., 2022*). The subcortical regions are also relatively small structures. For example, the STN has a volume of approximately 82 mm³; 3 mm isotropic resolutions therefore provide only three to four voxels for analysis of a relatively complex structure (*Alkemade et al., 2020*). Attaining sufficient signal in the deep brain for accurate statistical analysis is particularly difficult at lower field strengths (*Murphy et al., 2007*). Even with ultra-high field MRI and optimized sequences, it is sometimes difficult to approximate the reliability of the signal we receive. Indeed, the mismatch of findings between fMRI studies seemingly examining the same task, begs the question of reliability. While studies using lower field strengths have found it difficult to consistently replicate activation profiles in the SST, higher field strengths have had yielded better success. The subcortical activation profiles of inhibition painted by the three optimized 7T studies (deHollander_7T, Isherwood_7T, and Miletic_7T) are very consistent. We can therefore suggest that fMRI of the subcortex at higher field strengths is at the very least *reliable*.

A second point of consideration is the haemodynamics of the BOLD response in the subcortex, and the validity of our modelling methods in this context. A fundamental assumption of fMRI is that BOLD activity is influenced by neural activity. Although multiple studies have at least partially confirmed the linearity of the relationship between the HRF and neural activity (*Liu et al., 2010*; *Logothetis et al., 2001*), these findings have almost exclusively focused on the cortex (*Kim and Ress, 2017*; *Taylor et al., 2018*). Few studies have attempted to characterize the effect of vasculature on the HRF in the deep brain, though differences between the cortex and subcortex have been found (*Duvernoy, 1999*; *Lewis et al., 2018*; *Tatu et al., 1998*; *Wall et al., 2009*). Subcortical BOLD responses appear to peak earlier than those observed in the cortex and the post-stimulus undershoot normally associated with the canonical HRF is not always seen (*Kim et al., 2022*). Physiological noise is also more of an issue in the subcortex due to its proximity to large vessels (*Singh et al., 2018*). The cardiac system produces artefacts due to changes in blood flow and physical pulsation of vessels (*Dagli et al., 1999*; *Krüger and Glover, 2001*), while the respiratory system produces artefacts due to arterial pressure

changes and effects on B0 (*Raj et al., 2001*; *Wise et al., 2004*). Due to all of this, accurately imaging the subcortex requires numerous technical considerations. Moreover, an increase in BOLD signal most likely reflects underlying neural activation, but a lack of observed BOLD signal (especially in the subcortex) in no way suggests a *lack* of underlying neuronal activation (*Lowe et al., 2000*). So, while these results do not provide evidence towards involvement of the indirect or hyperdirect pathways in successful response inhibition, they also do not provide direct evidence against it.

As well as attempting to disentangle the pathways associated with inhibitory control specifically, we are also observing pathways related to attention and signal detection. This could explain the lack of subcortical findings for the SS > GO contrasts, but not the abundance of activation seen in the FS > GO contrast. From both the voxel- and ROI-wise analyses, it is clear that the VTA and STN play a role in the mechanisms underlying *FS* trials. Investigation into the specific role of the VTA and dopaminergic system in response inhibition has led to conflicting results (*Aron et al., 2003*; *Boonstra et al., 2005*). Inhibition of the VTA has been seen to increase the number of premature responses in a five-choice serial RT task in rats (*Flores-Dourojeanni et al., 2021*), the VTA appears to degenerate in Parkinson's disease (*Alberico et al., 2015*), and is associated with reward uncertainty in response inhibition tasks (*Tennyson et al., 2018*). Feedback in the SST is inherent in its design; a failure to stop simultaneously triggers the realization that an error was made, without the need for explicit task feedback. Speculatively, a failure to stop could trigger nodes of the mesolimbic pathway in response to this action error, as the VTA is known to respond to reward prediction errors (*Bayer and Neuron, 2005*; *Eshel et al., 2015*; *Schultz et al., 1997*). For the STN, we know that this region responds to errors (*Cavanagh et al., 2014*; *Siegert et al., 2014*), which is likely due to exertion of additional motor inhibition afterwards (*Guan and Wessel, 2022*). Therefore, one explanation is that the FS > GO contrast reflects an error mechanism. This is perhaps why this is not the first aggregatory study that has found limited activation of basal ganglia regions when specifically looking at successful response inhibition (*Hung et al., 2018*; *Isherwood et al., 2021b*; *Zhang et al., 2017*).

While more direct measurements of neural activity (e.g., LFP recordings) may provide superior temporal resolution, studies that use them are often required to observe electrophysiological responses in clinical populations, due to the invasive nature of the method. How such findings can translate to the neurotypical population continues to be a complex topic. Similarly, much work on the role of cortico-basal-ganglia loop in response inhibition has come from animal studies. While we can acquire data that is excellent in terms of temporal and spatial resolution and number of trials, how well this translates to the general human population is uncertain. Furthermore, while the temporal resolution of fMRI is inherently slower, more efficient methods for acquiring analysing fast fMRI data, show promise for overcoming this apparent drawback of fMRI (*Lewis et al., 2016*; *Polimeni and Lewis, 2021*). The ability to extract relevant neuronal information from early phases of the BOLD response may help in identifying activation from overlapping processes in complex tasks such as this. Several studies have established nonlinear properties of the hemodynamic response that have significant implications for fast fMRI (*Miller et al., 2001*; *Vazquez and Noll, 1998*). Of course, we cannot rule out that the range of fMRI data analysed here lacks the temporal resolution or sensitivity to observe significant changes in subcortical activation during motor inhibition, engagement, and instantiation of inhibition. Instead, this study picks up on the role of the basal ganglia in error processing, something that more direct methods of measurement have thus far not focused on.

From our literature search, this is the first study to take advantage of an array of unprocessed open-access data. Now that it is becoming increasingly common for researchers to make such detailed data available, it is beneficial for the research field to move on from meta-analytical methods that use only summary measures of activation profiles. Although, it should be noted, that canonical methods of meta-analysis still provide advantages over more processing-intensive methods such as applied here. Firstly, on the scale of five datasets, this methodology was applied and completed relatively quickly, but standard meta-analyses can include tens if not hundreds of studies, a feat that would be difficult to manage for the method described here. Secondly, the sample reported here is somewhat biased, it includes only data that were openly accessible in full, it is likely there are many more studies that would be useful to answer the research question posited here. Simpler methods of meta-analyses, where only coordinates or summary measures are needed to aggregate data, benefit from having access to a much wider range of potential sources of data. What we have been able to do here, even with a limited number of datasets, is process all the data with the same set of criteria. We therefore

benefit from sets of extremely well-vetted behavioural and functional data, that can have all aspects of the datasets compared to one another. This allowed us to tightly control aspects of the preprocessing pipeline that can affect later analyses steps, such as distortion correction and smoother kernel sizes.

The consequences of spatial smoothing on statistical analyses are well known and can have huge effects on group or subject-level inferences (*Chen and Calhoun, 2018*; *Mikl et al., 2008*). Here, we have shown again the substantial effect smoothing can have on the conclusions drawn from task-specific GLMs. In the absence of a ground truth, we are not able to fully justify the use of either larger or smaller kernels to analyse such data. On the one hand, aberrantly large smoothing kernels could lead to false positives in activation profiles, due to bleeding of observed activation into surrounding tissues. Conversely, too little smoothing could lead to false negatives, missing some true activity in surrounding regions. While we cannot concretely validate either choice, it should be noted that there is lower spatial uncertainty in the subcortex compared to the cortex, due to the lower anatomical variability. False positives from smoothing spatially unmatched signal, are therefore more likely than false negatives. It may be more prudent for studies to use a range of smoothing kernels, to assess the robustness of their fMRI activation profiles. Based on the results of the smoothing comparison and the differences in optimal kernel sizes for each dataset, ROI analyses may offer superior statistical testing to that of voxel-wise methods as you do not introduce a loss of specificity. ROI-wise methods also have the added benefit of not needing to warp images from individual space to common or group templates, which may also introduce a loss of specificity when looking at smaller structures, such as those in the subcortex. However, ROI-wise methods are only as good as the predefined atlases used in the analysis. MRI may benefit from high spatial resolution in comparison to other neuroimaging methods, but there are still subpopulations of nuclei, such as those within the STN, that may have different roles in response inhibition and are not easily distinguishable (*Mosher et al., 2021*). As already discussed in detail, MRI is of course also disadvantaged by its poor temporal resolution, a dimension that hinders the ability to dissociate different mechanisms occurring during the course of a trial (e.g., attention, detection of salient events; *Benis et al., 2016*). Methodologies with enhanced temporal resolution, such as electro-encephalography, will also benefit from the wave of open-access data and can focus on research questions that MRI currently cannot, including disentangling the mixed cognitive processes underlying response inhibition.

This paper serves as a proof of concept for methods of meta-analysis that allow the unification of largely unprocessed or unreduced datasets and exemplifies the huge opportunities that open-access data sharing can bring to the research field. As more and more datasets are made publicly available, researchers will be able to perform meta-analyses not only on summary data, but datasets with a rich body of parameters and data points. Our results indicate that error processing is likely a large driver of subcortical activity, and that nodes of the *indirect* and *hyperdirect* pathways appear to respond to this non-motor inhibition process more than to motor inhibition itself. We do not find evidence for either pathways involvement in successful motor inhibition, which may be a consequence of the overlap of inhibition control, attention, signal detection, and error processing on sub-second timescales in this task. Adaptations of the classical SST are already being deployed and may aid in the disentangling of attention and signal detection in overall response inhibition (*Boecker et al., 2013*; *Bryden and Roesch, 2015*).

## Materials and methods

### Participants

This study combined data from five datasets, two acquired at 3T and three at 7T: *Aron_3T* (*Aron and Poldrack, 2006*), *Poldrack_3T* (*Poldrack et al., 2016*), *deHollander_7T* (*de Hollander et al., 2017*), *Isherwood_7T* (*Isherwood et al., 2023*), and *Miletic_7T* (*Miletić et al., 2020*). The number of participants and their relevant demographics for each dataset are as follows: *Aron_3T* – 14 participants (4 females; mean age 28.1 ± 4.1), *Poldrack_3T* – 130 participants (62 females; mean age 31 ± 8.7; age range 21–50), *deHollander_7T* – 20 participants (10 females; mean age 26 ± 2.6; age range 22–32), *Isherwood_7T* – 37 participants (20 females; mean age 26.3 ± 5.6; age range 19–39), and *Miletic_7T* – 17 participants (9 females; mean age 23.7 ± 3.2).

**Table 2.** The principal MR acquisition parameters of the functional scans for each dataset.

| Dataset | TR (ms) | TE (ms) | Voxel size (mm) | FOV (mm) | No. slices | GRAPPA |
|---|---|---|---|---|---|---|
| *Aron_3T* | 2000 | 30 | 3.125 × 3.125 × 4 | 200 × 200 × 132 | 33 | N/A |
| *Poldrack_3T* | 2000 | 30 | 3 × 3 × 4 | 192 × 192 × 136 | 34 | N/A |
| *deHollander_7T* | 2000 | 14 | 1.5 × 1.5 × 1.5 | 192 × 192 × 97 | 60 | 3 |
| *Isherwood_7T* | 1380 | 14 | 1.5 × 1.5 × 1.5 | 192 × 192 × 128 | 82 | 3 |
| *Miletic_7T* | 3000 | 14 | 1.6 × 1.6 × 1.6 | 192 × 192 × 112 | 70 | 3 |

## Scanning protocols

This section describes the MR acquisition procedure for each dataset. The main acquisition parameters of the functionals scans can be found in *Table 2*, with a detailed account of each dataset's structural and functional scans in the following paragraphs.

For the *Aron_3T* dataset, each participant was scanned on a Siemens Allegra 3T scanner. The session consisted of three functional runs of the SST and an anatomical T1w image. The functional data were collected using a single echo 2D-echo planar imaging (EPI) BOLD sequence (TR = 2000 ms; TE = 30 ms; voxel size = 3.125 × 3.125 × 4 mm; flip angle = 90°; FOV = 200 × 200 × 132 mm; matrix size = 64 × 64; slices = 33; phase-encoding direction = A >> P). A 1-mm isotropic T1w image was acquired during each session using the MPRAGE sequence (TR = 2300 ms; TE = 2.1 ms; matrix size = 192 × 192).

For the *Poldrack_3T* dataset, each participant was scanned on a Siemens Trio 3T scanner. The session consisted of one functional run of the SST and an anatomical T1w image. The functional data were collected using a single echo 2D-EPI BOLD sequence (TR = 2000 ms; TE = 30 ms; voxel size = 3 × 3 × 4 mm; flip angle = 90°; FOV = 192 × 192 × 136 mm; matrix size = 64 × 64; slices = 34; phase-encoding direction = A >> P). A 1-mm isotropic T1w image was acquired during each session using the MPRAGE sequence (TR = 1900 ms; TE = 2.26 ms; matrix size = 256 × 256).

For the *deHollander_7T* dataset, each participant was scanned on a Siemens MAGNETOM 7 Tesla (7T) scanner with a 32-channel head coil. The session consisted of three functional runs of the SST, B0 field map acquisition (TR = 1500 ms, $TE_1$=6 ms, $TE_2$ = 7.02 ms), and an anatomical T1w image. The functional data were collected using a single echo 2D-EPI BOLD sequence (TR = 2000 ms; TE = 14 ms; GRAPPA = 3; voxel size = 1.5 mm isotropic; partial Fourier = 6/8; flip angle = 60°; FOV = 192 × 192 × 97 mm; matrix size = 128 × 128; BW = 1446 Hz/Px; slices = 60; phase-encoding direction = A >> P; echo spacing = 0.8 ms). Each run had an acquisition time of 13:27 min, totalling 40:21 min of functional scanning. A 0.7-mm isotropic T1w image was acquired during each session using the MP2RAGE sequence (TR = 5000 ms; TE = 2.45 ms; inversions TI1 = 900 ms, TI2 = 2750 ms; flip angle 1 = 5°; flip angle 2 = 3°; *Marques et al., 2010*).

For the *Isherwood_7T* dataset, each participant was scanned on a Siemens MAGNETOM TERRA 7T scanner with a 32-channel head coil. The session consisted of two functional runs of the SST, top-up acquisition, and an anatomical T1w image. The functional data were collected using a single echo 2D-EPI BOLD sequence (TR = 1380 ms; TE = 14 ms; MB = 2; GRAPPA = 3; voxel size = 1.5 mm isotropic; partial Fourier = 6/8; flip angle = 60°; FOV = 192 × 192 × 128 mm; matrix size = 128 × 128; BW = 1446 Hz/Px; slices = 82; phase-encoding direction = A >> P; echo spacing = 0.8 ms). Each run had an acquisition time of 13:27 min, totalling 26:54 min of functional scanning. Subsequently to each run, five volumes of the same protocol with opposite phase-encoding direction (P >> A) were collected (top-up) for distortion correction. A 1-mm isotropic T1w image was acquired during each session using the MP2RAGE sequence (TR = 4300 ms; TE = 1.84 ms; inversions TI1 = 840 ms, TI2 = 2370 ms; flip angle 1 = 5°; flip angle 2 = 6°; *Marques et al., 2010*).

For the *Miletic_7T* dataset, each participant was scanned on a Siemens MAGNETOM 7T scanner with a 32-channel head coil. The session consisted of three functional runs of the SST, B0 field map acquisition (TR = 1500 ms, $TE_1$ = 6 ms, $TE_2$ = 7.02 ms), and an anatomical T1w image. The functional data were collected using a single echo 2D-EPI BOLD sequence (TR = 3000 ms; TE = 14 ms; GRAPPA = 3; voxel size = 1.6 mm isotropic; partial Fourier = 6/8; flip angle = 70°; FOV = 192 × 192 × 112 mm; matrix size = 120 × 120; BW = 1436 Hz/Px; slices = 70; phase-encoding direction = A >>

P; echo spacing = 0.8 ms). A 0.7-mm isotropic T1w image was acquired during each session using the MP2RAGE sequence (TR = 5000 ms; TE = 2.45 ms; inversions TI1 = 900 ms, TI2 = 2750 ms; flip angle 1 = 5°; flip angle 2 = 3°; *Marques et al., 2010*).

## Procedure and exclusions

Participants that were not accompanied by a T1w anatomical image were automatically excluded from the study as the image is required for registration during preprocessing. In addition, the behavioural data of each participant from each database were quality controlled on the basis of a specific set of exclusion criteria. These criteria are: (1) more than 10% go omissions across all functional runs; (2) a stopping accuracy of less than 35% or more than 65%; (3) a go-accuracy of less than 95%; (4) mean signal respond RTs that were longer on average than go RTs (inconsistent with the standard race model). Based on these criteria, no subjects were excluded from the *Aron_3T* dataset. Twenty-four subjects were excluded from the *Poldrack_3T* dataset (3 based on criterion 1, 9 on criterion 2, 11 on criterion 3, and 8 on criterion 4). Three subjects were excluded from the *deHollander_7T* dataset (2 based on criterion 1 and 1 on criterion 2). Five subjects were excluded from the *Isherwood_7T* dataset (2 based on criterion 1, 1 on criterion 2, and 2 on criterion 4). Two subjects were excluded from the *Miletic_7T* dataset (1 based on criterion 2 and 1 on criterion 4). Note that some participants in the *Poldrack_3T* study failed to meet multiple inclusion criteria. A further nine participants were excluded from the *Poldrack_3T* dataset due to a lack of T1w image or a lack of SST data. As the specific genders and ages of each participant in each dataset are not all available due to General Data Protection Regulations, we were unable to recalculate participant demographics after exclusions. The final number of participants in each dataset after screening is as follows: *Aron_3T*, 14 participants; *Poldrack_3T*, 97 participants; *deHollander_7T*, 17 participants; *Isherwood_7T*, 31 participants; *Miletic_7T*, 15 participants. Therefore, the analyses in this paper are based on stop-signal data from 5 datasets, 174 participants, and 293 runs.

## Stop-signal task

All datasets used a simple, two alternative choice stop-signal paradigm. This paradigm consists of two trial types, go trials, and stop trials. On each trial, an arrow is presented on the screen in either the left or right direction (the go stimulus). The participant presses the button corresponding to the direction of the arrow. On a subset of trials (25%), a stop signal appears shortly after go signal onset, indicating the participant should try to inhibit their movement and not respond in that trial. In the auditory SST, this stop signal is presented as a 'beep' sound. In the visual SST, this stop signal is presented as a change in visual stimulus; for example, in the *Isherwood_7T* dataset, the circle surrounding the arrow would change from white to red. The time between the presentation of the go stimulus and the stop signal is defined by the SSD. The SSD is adapted iteratively during the task. Generally, if the participant responds during a stop trial, the SSD is reduced by 50 ms on the next stop trial, meaning the stop signal will appear earlier in the next trial and it will be easier for the participant to inhibit their response. Conversely, if the participant stops successfully, the SSD will increase by 50 ms and the stop signal will appear *later* in the next trial. This method of SSD adaptation is known as a staircase procedure and ensures that each participant is able to inhibit their actions approximately 50% of the time. Task performance in this paradigm is characterized by the race model (*Logan and Cowan, 1984*). The model assumes a go process and a stop process race independently and whichever finishes first defines whether a participant responds or inhibits their actions. The go process is characterized by the

**Table 3.** Task details for the SST in each dataset.

| Dataset | Response modality | Type | Stop-signal duration (ms) | No. staircases | SSD range (ms) | Total no. trials | Stop trials (%) |
|---|---|---|---|---|---|---|---|
| *Aron_3T* | Hand, R | Auditory | 500 | 4 | 100–250 | 384 | 25 |
| *Poldrack_3T* | Hand, R | Auditory | 250 | 2 | 0–1000 | 128 | 25 |
| *deHollander_7T* | Hand, L/R | Auditory | 62 | 4 | 0–900 | 384 | 25 |
| *Isherwood_7T* | Hand, L/R | Visual | 300 | 1 | 50–900 | 200 | 25 |
| *Miletic_7T* | Hand, L/R | Auditory | 62 | 2 | 0–900 | 342 | 25 |

observable go RT, whereas the stop process is characterized by the latent SSRT, which is estimated based on the effects of the SSD throughout the task.

Although the SST employed in each dataset is similar, there are some differences which are detailed in *Table 3*. We note here the most important differences in design aspects of the SSTs, these include (1) Response modality, describing the manual response and whether left (L), right (R), or both (L/R) hands were used; (2) Type, describing whether the stop signal was auditory or visual; (3) Stop-signal duration, how long the auditory or visual stop signal was presented for; (4) Number of staircases, describing the number of staircases used to track the SSD of each participant during the task; (5) SSD range, describing the minimum and maximum values that the SSD could be during the task; (6) Total trial number, the number of trials each participant performed over all runs; (7) Stop trials, the percentage of overall trials that were stop trials (as opposed to go trials).

## Behavioural analyses

For all runs within each dataset, median RTs on *go* and *stop* trials, the mean SSD and proportion of *successful stops* were calculated. For each participant, the SSRT was calculated using the integration method, with replacement of go omissions (*Verbruggen et al., 2019*) estimated by integrating the RT distribution and calculating the point at which the integral equals p(respond|signal). The completion time of the stop process aligns with the *n*th RT, where *n* equals the number of RTs in the RT distribution of go trials multiplied by the probability of responding to a signal. Both frequentist and Bayesian analyses methods were used to calculate the correlation between mean SSRTs and median go RTs, as well as to test the statistical difference between median failed stop RTs and median go RTs.

## fMRIprep preprocessing pipeline

*fMRIPrep* was used to preprocess all acquired anatomical and functional data (*Esteban et al., 2019*; *Esteban et al., 2020*). The following two sections describe, in detail, the preprocessing steps that *fMRIPrep* performed on each dataset.

## Anatomical data preprocessing

A total of 1 T1-weighted (T1w) images was found within the input for each subject of each BIDS dataset. The T1-weighted (T1w) image was corrected for intensity non-uniformity with *N4BiasFieldCorrection* (*Tustison et al., 2010*), distributed with ANTs 2.3.3 (*Avants et al., 2008*, RRID:SCR_004757), and used as T1w reference throughout the workflow. The T1w reference was then skull-stripped with a Nipype implementation of the *antsBrainExtraction.sh* workflow (from ANTs), using OASIS30ANTs as target template. Brain tissue segmentation of cerebrospinal fluid (CSF), white matter (WM), and grey matter (GM) was performed on the brain-extracted T1w using *fast* (FSL 5.0.9, RRID:SCR_002823, *Zhang et al., 2001*). Brain surfaces were reconstructed using *recon-all* (FreeSurfer 6.0.1, RRID:SCR_001847, *Dale et al., 1999*), and the brain mask previously estimated was refined with a custom variation of the method to reconcile ANTs- and FreeSurfer-derived segmentations of the cortical GM of Mindboggle (RRID:SCR_002438, *Klein et al., 2017*). Volume-based spatial normalization to one standard space (MNI152NLin2009cAsym) was performed through nonlinear registration with *antsRegistration* (ANTs 2.3.3) using brain-extracted versions of both T1w reference and the T1w template. The following template was selected for spatial normalization: *ICBM 152 Nonlinear Asymmetrical template version 2009c* (*Fonov et al., 2009*, RRID:SCR_008796; TemplateFlow ID: MNI152NLin2009cAsym).

## Functional data preprocessing

For each of the BOLD runs per subject (across all datasets), the following preprocessing was performed. First, a reference volume and its skull-stripped version were generated using a custom methodology of *fMRIPrep*. For datasets where a distortion correction image was not acquired (*Aron_3T* and *Poldrack_3T*), a deformation field to correct for susceptibility distortions was estimated based on *fMRIPrep*'s *fieldmap-less* approach. The deformation field is that resulting from co-registering the BOLD reference to the same-subject T1w reference with its intensity inverted (*Wang et al., 2017*). Registration is performed with *antsRegistration* (ANTs 2.3.3), and the process regularized by constraining deformation to be nonzero along the phase-encoding direction, and modulated with an average fieldmap template (*Treiber et al., 2016*). For the *deHollander_7T* and *Miletic_7T* datasets, a B0-nonuniformity map (or *fieldmap*) was estimated based on a phase-difference map calculated with

a dual-echo gradient-recall echo sequence, processed with a custom workflow of *SDCFlows* inspired by the *epidewarp.fsl* script with further improvements in HCP Pipelines (*Uğurbil et al., 2013*). The *fieldmap* was then co-registered to the target EPI reference run and converted to a displacements field map (amenable to registration tools such as ANTs) with FSL's fugue and other *SDCflows* tools. For the *Isherwood_7T* dataset, a B0-nonuniformity map (or *fieldmap*) was estimated based on two EPI references with opposing phase-encoding directions, with *3dQwarp* (*Cox and Hyde, 1997*; AFNI 20160207).

Based on the estimated susceptibility distortion, a corrected EPI reference was calculated for a more accurate co-registration with the anatomical reference. The BOLD reference was then co-registered to the T1w reference using bbregister (FreeSurfer) which implements boundary-based registration (*Greve and Fischl, 2009*). Co-registration was configured with six degrees of freedom. Head-motion parameters with respect to the BOLD reference (transformation matrices, and six corresponding rotation and translation parameters) were estimated before any spatiotemporal filtering using *mcflirt* (FSL 5.0.9, *Jenkinson et al., 2002*). BOLD runs were slice-time corrected using *3dTshift* from AFNI 20160207 (*Cox and Hyde, 1997*; RRID:SCR_005927). The BOLD timeseries (including slice-timing correction when applied) were resampled onto their original, native space by applying a single, composite transform to correct for head-motion and susceptibility distortions. These resampled BOLD timeseries will be referred to as *preprocessed BOLD in original space*, or just *preprocessed BOLD*. Several confounding timeseries were calculated based on the *preprocessed BOLD*: framewise displacement (FD), DVARS, and three region-wise global signals. FD was computed using two formulations following Power (absolute sum of relative motions, *Power et al., 2014*) and Jenkinson (relative root mean square displacement between affines, *Jenkinson et al., 2002*). FD and DVARS are calculated for each functional run, both using their implementations in *Nipype* (following the definitions by *Power et al., 2014*). The three global signals are extracted within the CSF, the WM, and the whole-brain masks. Additionally, a set of physiological regressors were extracted to allow for component-based noise correction (*Behzadi et al., 2007*). Principal components are estimated after high-pass filtering the *preprocessed BOLD* timeseries (using a discrete cosine filter with 128 s cut-off) for the two *CompCor* variants: temporal (tCompCor) and anatomical (aCompCor). tCompCor components are then calculated from the top 2% variable voxels within the brain mask. For aCompCor, three probabilistic masks (CSF, WM, and combined CSF + WM) are generated in anatomical space. The implementation differs from that of Behzadi et al. in that instead of eroding the masks by 2 pixels on BOLD space, the aCompCor masks are subtracted a mask of pixels that likely contain a volume fraction of GM. This mask is obtained by dilating a GM mask extracted from the FreeSurfer's *aseg* segmentation, and it ensures components are not extracted from voxels containing a minimal fraction of GM. Finally, these masks are resampled into BOLD space and binarized by thresholding at 0.99 (as in the original implementation). Components are also calculated separately within the WM and CSF masks. For each CompCor decomposition, the *k* components with the largest singular values are retained, such that the retained components' timeseries are sufficient to explain 50% of variance across the nuisance mask (CSF, WM, combined, or temporal). The remaining components are dropped from consideration. The head-motion estimates calculated in the correction step were also placed within the corresponding confounds file. The confound timeseries derived from head-motion estimates and global signals were expanded with the inclusion of temporal derivatives and quadratic terms for each (*Satterthwaite et al., 2013*). Frames that exceeded a threshold of .5 mm FD or 1.5 standardized DVARS were annotated as motion outliers. All resamplings can be performed with *a single interpolation step* by composing all the pertinent transformations (i.e., head-motion transform matrices, susceptibility distortion correction when available, and co-registrations to anatomical and output spaces). Gridded (volumetric) resamplings were performed using *antsApplyTransforms* (ANTs), configured with Lanczos interpolation to minimize the smoothing effects of other kernels (*Lanczos, 1964*). Non-gridded (surface) resamplings were performed using *mri_vol2surf* (FreeSurfer).

## Temporal signal-to-noise ratios

Sequence sensitivity in BOLD fMRI can be approximated by the calculation of the tSNR. While it is not possible discriminate the exact source of noise causing temporal fluctuations in measured signal, they are thought to arise from either thermal or physiological interference. To get a feel for the image quality in different regions of the brain between datasets, we here compared ROI-wise tSNRs. Using

probabilistic atlases, we took the mean of the ROI signal and divided by its standard deviation across time. Each voxels contribution to the mean signal of the region was weighted by its probability of belonging to the region While simple to calculate, tSNR comparison between data of differing acquisition methods is less trivial. Here, we only correct for the differences in voxel size between datasets. As spatial resolution is directly proportional to MR signal, we divided these tSNR values by the volume of a single voxel (*Edelstein et al., 1986*). tSNR was calculated using the exact same data used in the ROI-wise GLMs. That is, unsmoothed but preprocessed data from fMRIprep.

## fMRI analysis – GLMs

GLM analyses were computed at both a whole-brain voxel-wise and region-specific level. A canonical double gamma HRF with temporal derivative was used as the basis set for both methods of analysis (*Glover, 1999*). The design matrix consisted of the three task-specific regressors for each of the three experimental conditions: failed stop (FS) trials, successful stop (SS) trials, and go (GO) trials, six motion parameters (three translational and three rotational) as well as DVARS and FD estimated during preprocessing. The first 20 aCompCor components from *fMRIPrep* were used to account for physiological noise (*Behzadi et al., 2007*). For the main GLM analyses, all events were time-locked to the GO signal onset, but see the supplementary analyses (*Figure 2—figure supplement 4*, *Figure 3—figure supplement 1*) for results where SS and FS trials were time-locked to the stop-signal onset. Following data preprocessing through *fMRIPrep*, all data were high-pass filtered (cut-off 1/128 Hz) to remove slow drift. Three SST contrasts were computed for both the whole-brain and ROI GLMs: *FS > GO*, *FS > SS*, and *SS > GO*. While many regressors were computed in the preprocessing of the fMRI data, not all were used in the subsequent analysis. The exact regressors used for the analysis can be found above. For example, tCompCor and global signals were calculated in our generic preprocessing pipeline but not part of the analysis. The code used for preprocessing and analysis can be found in the data and code availability statement.

## Voxel-wise

Whole-brain analyses were computed using the FILM method from FSL FEAT (version 6.0.5.2; *Jenkinson et al., 2012*; *Woolrich et al., 2001*) as implemented in the Python package wrapper Nipype (version 1.7.0; *Gorgolewski et al., 2011*). Run-level GLMs accounting for autocorrelated residuals were computed, the results warped to MNI152NLin2009cAsym space, and subsequently combined per subject using fixed effects analyses. Data for the whole-brain GLMs were spatially smoothed using the SUSAN method with a full width half maximum (FWHM) equal to the voxel size of the functional image (*Smith and Brady, 1997*). Therefore, a 3.125-mm kernel was applied to the *Aron_3T* dataset, a 3-mm kernel to the *Poldrack_3T* dataset, a 1.5-mm kernel to the *deHollander_7T* and *Isherwood_7T* datasets, and a 1.6-mm kernel to the *Miletic_7T* dataset. These base-level kernels were applied to the data used for the main statistical analyses. Group-level models were subsequently estimated using FMRIB Local Analysis of Mixed Effects (FLAME) 1 and FLAME 2 from FSL (*Woolrich et al., 2001*), taking advantage of the fact that FLAME allows the estimation of different variances for each dataset. Dummy variables were used as regressors to allow the categorization of data into different datasets so that they could be estimated separately and then combined. SPMs were generated to visualize the resulting group-level models. The maps were corrected for the FDR using critical value of $q < 0.05$ (*Yekutieli and Benjamini, 1999*).

## ROI-wise

ROI analyses were then performed. Timeseries were extracted from each subcortical ROI using probabilistic masks provided by MASSP (*Bazin et al., 2020*), except in the case of the putamen and caudate nucleus, which were provided by the Harvard-Oxford subcortical atlas (*Rizk-Jackson et al., 2011*). Each voxels contribution to the mean signal of the region was therefore weighted by its probability of belonging to the region. Cortical regions parcellations were provided by the Harvard-Oxford cortical atlas (*Rizk-Jackson et al., 2011*). These timeseries were extracted from unsmoothed data so to ensure regional specificity. ROI analyses were computed using the FILM method of FSL FEAT. To do this, we inputted each run for each participant in MNI152NLin2009cAsym space, where the signal of each region was replaced with its mean extracted timeseries. Hence, the signal within each region was homogenous on each given volume. Note that the standard implementation of FSL FILM

**Table 4.** Approximate interpretation of logarithmically transformed Bayes Factors. H1 represents the alternative hypothesis, H0 represents the null hypothesis.

| Log$_{10}$ BF | | | Interpretation |
|---|---|---|---|
| | > | 2 | Extreme evidence for H1 |
| 1.5 | – | 2 | Very strong evidence for H1 |
| 1 | – | 1.5 | Strong evidence for H1 |
| 0.5 | – | 1 | Moderate evidence for H1 |
| 0 | – | 0.5 | Anecdotal evidence for H1 |
| | 0 | | No evidence |
| 0 | – | −0.5 | Anecdotal evidence for H0 |
| −0.5 | – | −1 | Moderate evidence for H0 |
| −1 | – | −1.5 | Strong evidence for H0 |
| −1.5 | – | −2 | Very strong evidence for H0 |
| −2 | > | | Extreme evidence for H0 |

uses a spatial smoothing procedure prior to estimating temporal autocorrelations which is suitable for use only on voxel-wise data (*Woolrich et al., 2001*). We therefore turned this spatial smoothing procedure off and instead estimated autocorrelation using each voxel's individual timeseries. ROIs were therefore defined before implementing the ROI analyses. The regions include the IFG, primary motor cortex (M1), preSMA, caudate nucleus (caudate), GPe, GPi, putamen, SN, STN, Tha, and VTA. Due to the restricted FOV of the *deHollander_7T* dataset, this dataset was not used in the ROI-wise analysis of the M1 and preSMA regions. M1 and preSMA ROI-wise results are therefore based only on the *Aron_3T*, *Poldrack_3T*, *Isherwood_7T*, and *Miletic_7T* datasets. After the run-level GLMs were computed using FILM, the same fixed effects analyses and subsequent mixed-effects analyses used in the voxel-wise GLMs were performed. In addition to the frequentist analysis, we computed BFs for each contrast per ROI and hemisphere. To do this, we extracted the beta weights for each individual trial type from our first-level model. We then compared the beta weights from each trial type to one another using the 'BayesFactor' package as implemented in R (*Morey and Rouder, 2015*).

We compared the full model (H1) comprising trial type, dataset and subject as predictors to the null model (H0) comprising only the dataset and subject as predictor. Datasets and subjects were modelled as random factors in both cases. Since effect sizes in fMRI analyses are typically small, we set the scaling parameter on the effect size prior for fixed effects to 0.25, instead of the default of 0.5, which assumes medium effect sizes (note that the same qualitative conclusions would be reached with the default prior setting; *Rouder et al., 2009*). We calculated the BF for the full model over the null model, to provide evidence for or against a difference in beta weights for each trial type. To interpret the BFs, we used a modified version of Jeffreys' scale (*Andraszewicz et al., 2015*; *Jeffreys, 1939*). To facilitate interpretation of the BFs, we converted them to the logarithmic scale. The approximate conversion between the interpretation of logarithmic BFs and standard interpretation on the adjusted Jeffreys' scale can be found in *Table 4*.

## tSNRs
### Smoothing comparison
To further understand the impact of preprocessing on fMRI analyses, we computed voxel-wise GLM results based on a more lenient smoothing kernel. To observe the effect of smoothing on these analyses, we compared the results of our main statistical analyses, using base-level kernel sizes, to the same data when all datasets were smoothed using a 5-mm FWHM kernel. We chose to compared base-level smoothing kernels to 5 mm as this was the kernel sized used in the *Aron and Poldrack, 2006* study. To do this, the same voxel-wise GLM method was used as described above.

## Acknowledgements

The authors would like to thank Russ Poldrack for assistance with the analysis of the 3T datasets, and Alex Sebastian for thoughtful discussions during the conceptualization of the project.

## Additional information

### Funding

| Funder | Grant reference number | Author |
|--------|------------------------|--------|
| Nederlandse Organisatie voor Wetenschappelijk Onderzoek | ERC-2020-Cog-864750 | Birte Forstmann |

The funders had no role in study design, data collection and interpretation, or the decision to submit the work for publication.

### Author contributions

Scott Isherwood, Conceptualization, Data curation, Software, Formal analysis, Validation, Investigation, Visualization, Methodology, Writing – original draft, Project administration, Writing – review and editing; Sarah A Kemp, Conceptualization, Data curation, Software, Validation, Methodology, Writing – original draft, Project administration, Writing – review and editing; Steven Miletić, Conceptualization, Data curation, Software, Formal analysis, Validation, Investigation, Methodology, Writing – review and editing; Niek Stevenson, Conceptualization, Validation, Methodology, Writing – review and editing; Pierre-Louis Bazin, Conceptualization, Data curation, Software, Supervision, Validation, Methodology, Writing – review and editing; Birte Forstmann, Conceptualization, Supervision, Funding acquisition, Writing – review and editing

### Author ORCIDs

Scott Isherwood (D) https://orcid.org/0000-0003-2045-9268
Steven Miletić (D) https://orcid.org/0000-0001-7399-2926

Reviewer #2 (Public review): https://doi.org/10.7554/eLife.88652.4.sa1
Author response https://doi.org/10.7554/eLife.88652.4.sa2

## Additional files

### Supplementary files
MDAR checklist

### Data availability
Data were aggregated from five extant, open-source fMRI datasets. All analysis code can be found at https://osf.io/3h6rc/.

The following dataset was generated:

| Author(s) | Year | Dataset title | Dataset URL | Database and Identifier |
|-----------|------|---------------|-------------|-------------------------|
| Kemp SA, Miletic S, Isherwood S, Stevenson N, Bazin PL, Forstmann B | 2024 | Multi-study fMRI outlooks on subcortical BOLD responses in the stop-signal paradigm | https://osf.io/3h6rc/ | Open Science Framework, 3h6rc |

The following previously published datasets were used:

| Author(s) | Year | Dataset title | Dataset URL | Database and Identifier |
|---|---|---|---|---|
| Aron AR, Behrens TE, Frank M, Smith S, Poldrack RA | 2016 | Stop-signal task with unconditional and conditional stopping | https://openneuro.org/datasets/ds000008/versions/00001 | OpenNeuro, ds000008 |
| Sabb F, Karlsgodt K, Congdon E, Freimer N, London E, Cannon T, Poldrack R, Bilder R | 2016 | UCLA Consortium for Neuropsychiatric Phenomics LA5c Study | https://openfmri.org/dataset/ds000030/ | OpenfMRI, ds000030 |
| Isherwood SJS, Bazin PLEA, Miletić S, Stevenson NR, Trutti AC, Tse DHY, Heathcote A, Matzke D, Habli S, Sokołowski DR, Alkemade A, Håberg A, Forstmann B | 2023 | Investigating Intra-Individual Networks of Response Inhibition and Interference Resolution using 7T MRI data | https://doi.org/10.21942/uva.22240393.v1 | figshare, 10.21942/uva.22240393.v1 |
| Miletić S, Bazin PL, Weiskopf N, van der Zwaag W, Forstmann BU, Trampel R | 2020 | MRI protocol optimization for simultaneously studying small subcortical and cortical areas at 7 T | https://osf.io/6qwjz/ | Open Science Framework, 6qwjz |
| de Hollander G, Keuken M, Cvan der Zwaag W, Forstmann BU, Trampel R | 2023 | The canonical stopping network: Revisiting the role of the subcortex in response inhibition data | https://doi.org/10.21942/uva.23579172.v1 | figshare, 10.21942/uva.23579172.v1 |

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
