## [Editor Report · eLife Assessment]

This study aggregates across five fMRI datasets and reports that a network of brain areas previously associated with response inhibition processes, including several in the basal ganglia, are more active on failed stop than successful stop trials. This study is **valuable** as a well-powered investigation of fMRI measures of stopping, and following revisions provides **solid** evidence for its conclusions.

---

## [Referee Report · Reviewer #2 (Public review)]

This work aggregates data across 5 openly available stopping studies (3 at 7 tesla and 2 at 3 tesla) to evaluate activity patterns across the common contrasts of Failed Stop (FS) > Go, FS > stop success (SS), and SS > Go. Previous work has implicated a set of regions that tend to be positively active in one or more of these contrasts, including the bilateral inferior frontal gyrus, preSMA, and multiple basal ganglia structures. However, the authors argue that upon closer examination, many previous papers have not found subcortical structures to be more active on SS than FS trials, bringing into question whether they play an essential role in (successful) inhibition. In order to evaluate this with more data and power, the authors aggregate across five datasets and find many areas that are *more* active for FS than SS, including bilateral preSMA, GPE, thalamus, and VTA. They argue that this brings into question the role of these areas in inhibition, based upon the assumption that areas involved in inhibition should be more active on successful stop than failed stop trials, not the opposite as they observed.

Comments on revisions:

The authors have been responsive to the feedback of both reviewers and they have significantly improved the manuscript. I now judge the work as valuable and solid. The authors have achieved their aims to characterize subcortical BOLD activation in the stop-signal paradigm.

---

## [Author Response]

The following is the authors’ response to the previous reviews.

**Reviewer 1:**
This study is one in a series of excellent papers by the Forstmann group focusing on the ability of fMRI to reliably detect activity in small subcortical nuclei - in this case, specifically those purportedly involved in the hyper- and indirect inhibitory basal ganglia pathways. I have been very fond of this work for a long time, beginning with the demonstration of De Hollander, Forstmann et al. (HBM 2017) of the fact that 3T fMRI imaging (as well as many 7T imaging sequences) do not afford sufficient signal to noise ratio to reliably image these small subcortical nuclei. This work has done a lot to reshape my view of seminal past studies of subcortical activity during inhibitory control, including some that have several thousand citations.Comments on revised version:This is my second review of this article, now entitled "Multi-study fMRI outlooks on subcortical BOLD responses in the stop-signal paradigm" by Isherwood and colleagues.The authors have been very responsive to the initial round of reviews.I still think it would be helpful to see a combined investigation of the available 7T data, just to really drive the point home that even with the best parameters and a multi-study sample size, fMRI cannot detect any increases in BOLD activity on successful stop compared to go trials. However, I agree with the authors that these "sub samples still lack the temporal resolution seemingly required for looking at the processes in the SST." As such, I don't have any more feedback.

We thank the reviewer for their positive feedback, and for their thorough and constructive comments on our initial submission.

**Reviewer 2:**
This work aggregates data across 5 openly available stopping studies (3 at 7 tesla and 2 at 3 tesla) to evaluate activity patterns across the common contrasts of Failed Stop (FS) > Go, FS > stop success (SS), and SS > Go. Previous work has implicated a set of regions that tend to be positively active in one or more of these contrasts, including the bilateral inferior frontal gyrus, preSMA, and multiple basal ganglia structures. However, the authors argue that upon closer examination, many previous papers have not found subcortical structures to be more active on SS than FS trials, bringing into question whether they play an essential role in (successful) inhibition. In order to evaluate this with more data and power, the authors aggregate across five datasets and find many areas that are *more* active for FS than SS, including bilateral preSMA, GPE, thalamus, and VTA. They argue that this brings into question the role of these areas in inhibition, based upon the assumption that areas involved in inhibition should be more active on successful stop than failed stop trials, not the opposite as they observed.Since the initial submission, the authors have improved their theoretical synthesis and changed their SSRT calculation method to the more appropriate integration method with replacement for go omissions. They have also done a better job of explaining how these fMRI results situate within the broader response inhibition literature including work using other neuroscience methods.They have also included a new Bayes Factor analysis. In the process of evaluating this new analysis, I recognized the following comments that I believe justify additional analyses and discussion:First, if I understand the author's pipeline, for the ROI analyses it is not appropriate to run FSL's FILM method on the data that were generated by repeating the same time series across all voxels of an ROI. FSL's FILM uses neighboring voxels in parts of the estimation to stabilize temporal correlation and variance estimates and was intended and evaluated for use on voxelwise data. Instead, I believe it would be more appropriate to average the level 1 contrast estimates over the voxels of each ROI to serve as the dependent variables in the ROI analysis.

We agree with the reviewer’s assertion that this approach could create estimation problems. However, in this instance, we turned off the spatial smoothing procedure that FSL’s FILM normally uses for estimating the amount of autocorrelation – thus, the autocorrelation was estimated based on each voxel’s timeseries individually. We also confirmed that all voxels within each ROI had identical statistics, which would not be the case if the autocorrelation estimates differed per voxel. We have added the following text to the Methods section under fMRI analysis: ROI-wise:

Note that the standard implementation of FSL FILM uses a spatial smoothing procedure prior to estimating temporal autocorrelations which is suitable for use only on voxelwise data (Woolrich et al., 2001). We therefore turned this spatial smoothing procedure off and instead estimated autocorrelation using each voxel’s individual timeseries.

Second, for the group-level ROI analyses there seems to be inconsistencies when comparing the z-statistics (Figure 3) to the Bayes Factors (Figure 4) in that very similar zstatistics have very different Bayes Factors within the same contrast across different brain areas, which seemed surprising (e.g., a z of 6.64 has a BF of .858 while another with a z of 6.76 has a BF of 3.18). The authors do briefly discuss some instances in the frequentist and Bayesian results differ, but they do not ever explain by similar z-stats yield very different bayes factors for a given contrast across different brain areas. I believe a discussion of this would be useful.

We thank the reviewer for their keen observation, and agree that this is indeed a strange inconsistency. Upon reviewing this issue, we came across an error in our analysis pipeline, which led to inconsistent scaling of the parameter estimates between datasets. We corrected this error, and included new tables (Figures 3, 4, and Supplementary Figure 5) which now show improved correspondence between the frequentist results from FSL and the Bayesian results.

We have updated the text of the Results section accordingly. In this revision, we have also updated all BFs to be expressed in log_10_ form, to ensure consistency for the reader. Updates to the manuscript are given below.

Results: Behavioural Analyses:

Consistent with the assumptions of the standard horse-race model (Logan & Cowan, 1984), the median failed stop RT is significantly faster within all datasets than the median go RT (Aron_3T: p < .001, BF_log10_ = 2.77; Poldrack_3T: p < .001, BF_log10_ = 23.49; deHollander_7T: p < .001, B BF_log10_ = 8.88; Isherwood_7T: p < .001, BF_log10_ = 2.95; Miletic_7T: p = .0019, BF_log10_ = 1.35). Mean SSRTs were calculated using the integration method and are all within normal range across the datasets.

Results: ROI-wise GLMS:

To further statistically compare the functional results between datasets, we then fit a set of GLMs using the canonical HRF with a temporal derivative to the timeseries extracted from each ROI. Below we show the results of the group-level ROI analyses over all datasets using z-scores (Fig. 3) and log-transformed Bayes Factors (BF; Fig. 4). Note that these values were time-locked to the onset of the go signal. See Supplementary Figure 5 for analyses where the FS and SS trials were time-locked to the onset of the stop signal. To account for multiple comparisons, threshold values were set using the FDR method for the frequentist analyses.

For the FS > GO contrast, the frequentist analysis found significant positive z-scores in all regions bar left and right M1, and the left GPi. The right M1 showed a significant negative z-score; left M1 and GPi showed no significant effect in this contrast. The BFs showed moderate or greater evidence for the alternative hypothesis in bilateral IFG, preSMA, caudate, STN, Tha, and VTA, and right GPe. Bilateral M1 and left GPi showed moderate evidence for the null. Evidence for other ROIs was anecdotal (see Fig 4).

For the FS > SS contrast, we found significant positive z-scores in in all regions except the left GPi. The BFs showed moderate or greater evidence for right IFG, right GPi, and bilateral M1, preSMA, Tha, and VTA, and moderate evidence for the null in left GPi. Evidence for other ROIs was anecdotal (see Fig 4).

For the SS > GO contrast we found a significant positive z-scores in bilateral IFG, right Tha, and right VTA, and significant negative z-scores in bilateral M1, left GPe, right GPi, and bilateral putamen. The BFs showed moderate or greater evidence for the alternative hypothesis in bilateral M1 and right IFG, and moderate or greater evidence for the null in left preSMA, bilateral caudate, bilateral GPe, left GPi, bilateral putamen, and bilateral SN. Evidence for other ROIs was anecdotal (see Fig 4).

Although the frequentist and Bayesian analyses are mostly in line with one another, there were also some differences, particularly in the contrasts with FS. In the FS > GO contrast, the interpretation of the GPi, GPe, putamen, and SN differ. The frequentist models suggests significantly increased activation for these regions (except left GPi) in FS trials. In the Bayesian model, this evidence was found to be anecdotal in the SN and right GPi, and moderate in the right GPe, while finding anecdotal or moderate evidence for the null hypothesis in the left GPe, left GPi, and putamen. For the FS > SS contrast, the frequentist analysis showed significant activation in all regions except for the left GPi, whereas the Bayesian analysis found this evidence to be only anecdotal, or in favour of the null for a large number of regions (see Fig 4 for details).

Since the Bayes Factor analysis appears to be based on repeated measures ANOVA and the z-statistics are from Flame1+2, the BayesFactor analysis model does not pair with the frequentist analysis model very cleanly. To facilitate comparison, I would recommend that the same repeated measures ANOVA model should be used in both cases. My reading of the literature is that there is no need to be concerned about any benefits of using Flame being lost, since heteroscedasticity does not impact type I errors and will only potentially impact power.

We agree with the reviewer that there are differences between the two analyses. The advantage of the z-statistics from FSL’s flame 1+2 is that these are based on a multi-level model in which measurement error in the first level (i.e., subject level) is taken into account in the group-level analysis. This is an advantage especially in the current paper since the datasets differ strongly in the degree of measurement error, both due to the differences in field strength and in the number of trials (and volumes). Although multilevel Bayesian approaches exist, none (except by use of custom code) allow for convolution with the HRF of a design matrix like typical MRI analyses. Thus, we extracted the participant-level parameter estimates (converted to percent signal change), and only estimated the dataset and group level parameters with the BayesFactor package. As such, this approach effectively ignores measurement error. However, despite these differences in the analyses, the general conclusions from the Bayesian and frequentist analyses are very aligned after we corrected for the error described above. The Bayesian results are more conservative, which can be explained by the unfiltered participantlevel measurement error increasing the uncertainty of the group-level parameter estimates. At worst, the BFs represent the lower bounds of the true effect, and are thus safe to interpret.

We have also included an additional figure (Supplementary Figure 7) that shows the correspondence between the BFs and the z scores.

Though frequentist statistics suggest that many basal ganglia structures are significantly more active in the FS > SS contrast (see 2nd row of Figure 3), the Bayesian analyses are much more equivocal, with no basal ganglia areas showing Log10BF > 1 (which would be indicative of strong evidence). The authors suggest that "the frequentist and Bayesian analyses are monst in line with one another", but in my view, this frequentist vs. Bayesian analysis for the FS > SS contrast seems to suggest substantially different conclusions. More specifically, the frequentist analyses suggest greater activity in FS than SS in most basal ganglia ROIs (all but 2), but the Bayesian analysis did not find *any* basal ganglia ROIs with strong evidence for the alternative hypothesis (or a difference), and several with more evidence for the null than the alternative hypothesis. This difference between the frequentist and Bayesian analyses seems to warrant discussion, but unless I overlooked it, the Bayesian analyses are not mentioned in the Discussion at all. In my view, the frequentist analyses are treated as the results, and the Bayesian analyses were largely ignored.

The original manuscript only used frequentist statistics to assess the results, and then added Bayesian analyses later in response to a reviewer comment. We agree that the revised discussion did not consider the Bayesian results in enough detail, and have updated the manuscript throughout to more thoroughly incorporate the Bayesian analyses and improve overall readability.

In the Methods section, we have updated the fMRI analysis – general linear models (GLMs): ROIwise GLMs section to more thoroughly incorporate the Bayesian analyses as follows:

We compared the full model (H1) comprising trial type, dataset and subject as predictors to the null model (H0) comprising only the dataset and subject as predictor. Datasets and subjects were modeled as random factors in both cases. Since effect sizes in fMRI analyses are typically small, we set the scaling parameter on the effect size prior for fixed effects to 0.25, instead of the default of 0.5, which assumes medium effect sizes (note that the same qualitative conclusions would be reached with the default prior setting; Rouder et al., 2009). We divided the resultant BFs from the full model by the null model to provide evidence for or against a difference in beta weights for each trial type. To interpret the BFs, we used a modified version of Jeffreys’ scale (Andraszewicz et al., 2014; Jeffreys, 1939). To facilitate interpretation of the BFs, we converted them to the logarithmic scale. The approximate conversion between the interpretation of logarithmic BFs and standard interpretation on the adjusted Jeffreys’ scale can be found in Table 4.

The Bayesian results are also more incorporated into the Discussion as follows:

Evidence for the role of the basal ganglia in response inhibition comes from a multitude of studies citing significant activation of either the SN, STN or GPe during successful inhibition trials (Aron, 2007; Aron & Poldrack, 2006; Mallet et al., 2016; Nambu et al., 2002; Zhang & Iwaki, 2019). Here, we re-examined activation patterns in the subcortex across five different datasets, identifying differences in regional activation using both frequentist and Bayesian approaches. Broadly, the frequentist approach found significant differences between most ROIs in FS>GO and FS>SS contrasts, and limited differences in the SS>GO contrast. The Bayesian results were more conservative; while many of the ROIs showed moderate or strong evidence, some with small but significant z scores were considered only anecdotal by the Bayesian analysis. In our discussion, where the findings between analytical approaches differ, we focus mainly on the more conservative Bayesian analysis.

Here, our multi-study results found limited evidence that the canonical inhibition pathways (the indirect and hyperdirect pathways) are recruited during successful response inhibition in the SST. We expected to find increased activation in the nodes of the indirect pathway (e.g., the preSMA, GPe, STN, SN, GPi, and thalamus) during successful stop compared to go or failed stop trials. We found strong evidence for activation pattern differences in the preSMA, thalamus, and right GPi between the two stop types (failed and successful), and limited evidence, or evidence in favour of the null hypothesis, in the other regions, such as the GPe, STN, and SN. However, we did find recruitment of subcortical nodes (VTA, thalamus, STN, and caudate), as well as preSMA and IFG activation during failed stop trials. We suggest that these results indicate that failing to inhibit one’s action is a larger driver of the utilisation of these nodes than action cancellation itself.

These results are in contention to many previous fMRI studies of the stop signal task as well as research using other measurement techniques such as local field potential recordings, direct subcortical stimulation, and animal studies, where activation of particularly the STN has consistently been observed (Alegre et al., 2013b; Aron & Poldrack, 2006; Benis et al., 2014; Fischer et al., 2017; Mancini et al., 2019; Wessel et al., 2016).